

# High Affinity Tamoxifen Analogues Retain Extensive Positional Disorder when Bound to Calmodulin

Lilia Milanesi[1,2], Clare R. Trevitt[1], Brian Whitehead[1], Andrea M. Hounslow[1], Salvador Tomas[2], Laszlo L. P. Hosszu[1,3], Christopher A. Hunter[4], and Jonathan P. Waltho[1,5]

[1]Department of Molecular Biology and Biotechnology, University of Sheffield, Sheffield S10 2TN, UK
[2]Department of Biological Sciences, School of Science, Birkbeck University of London, London WC1E 7HX, UK
[3]Medical Research Council Prion Unit, University College of London Institute of Neurology, Queen Square, London WCN1 3BG, UK
[4]Department of Chemistry, University of Cambridge, Lensfield Road, Cambridge CB2 1EW, UK
[5]Manchester Institute of Biotechnology, University of Manchester, 131 Princess St, Manchester, M1 7DN, UK

*Correspondence to*: Jonathan Waltho (j.waltho@sheffield.ac.uk)

Keywords Protein NMR, calmodulin, tamoxifen, ligand dynamics, protein-ligand complex

**Abstract.** Using a combination of NMR and fluorescence measurements we have investigated the structure and dynamics of the complexes formed between calcium loaded calmodulin (CaM) and the potent breast cancer inhibitor idoxifene, a derivative

of tamoxifen. High affinity binding ($K_d \sim 300$ nM) saturates with a 2:1 idoxifene:CaM complex. The complex is an ensemble where each idoxifene molecule is predominantly in the vicinity of one of the two hydrophobic patches of CaM but, in contrast with the lower affinity antagonists TFP, J-8 and W-7, does not substantially occupy the hydrophobic pocket. At least four idoxifene orientations per domain of CaM are necessary to satisfy the intermolecular NOE restraints, and this requires that the idoxifene molecules switch rapidly between positions. The CaM molecule is predominantly in the form where the N and C-

terminal domains are in close proximity allowing for the idoxifene molecules to contact both domains simultaneously. Hence, the 2:1 idoxifene:CaM complex illustrates how high affinity binding occurs without the loss of extensive positional dynamics.

## 1 Introduction

Calmodulin (CaM) is an important intracellular calcium receptor found in all eukaryotic cells. Calcium loaded CaM binds to more than 300 target enzymes that modulate various cellular functions (Ikura and Ames, 2006; Swulius and Waxham, 2008).

CaM consists of two globular domains separated by a solvent exposed helical region that is not continuous in solution (the so-called flexible tether), allowing the two domains to be independently mobile (Barbato et al., 1992; Chou et al., 2001; Trevitt et al., 2005). On binding calcium, the four helices in each of the two globular domains undergo a large conformational change, where the domains become less compact and a hydrophobic pocket is opened (Finn et al., 1995; Kuboniwa et al., 1995; Zhang et al., 1995). These hydrophobic pockets play a central role in the binding of various CaM targets (Meador et al., 1992; Ikura

et al., 1992; Craven et al., 1996; Osawa et al., 1998; Harmat et al., 2000; Kovesi et al., 2008).



The proposed mechanism of action of CaM mediated enzyme regulation is based largely on structural studies of complexes between CaM and peptides of 20-30 residues corresponding to CaM interaction domains, rather than intact enzymes. On binding of most of these peptides to CaM, the flexible tether between the two globular domains bends such that the N-terminal domain comes close to the C-terminal domain and the α-helices that usually form in the bound peptides stabilize and fix the

position of the two CaM domains (Maximciuc et al., 2006; Frederick et al., 2007). In this mode of binding, also called the wrap-around mode, the hydrophobic pockets in the globular domains become occupied by sidechains of hydrophobic residues within the peptides and the complex adopts a compact, globular structure. However, more recent structural studies show alternative modes of CaM binding to proteins and peptides. In some of these complexes, CaM adopts an extended structure more similar to that of uncomplexed CaM (Elshorst et al., 1999; Samal et al., 2011) and the hydrophobic pockets of CaM do

not bind to the hydrophobic residues of the target peptide, although nanomolar binding affinity is retained (Yamauchi et al., 2003; Izumi et al., 2008).

A similar diversity of binding modes has been observed in CaM bound to small molecule antagonists that share features of the target peptides. They have hydrophobic regions and basic functional groups but have greater mobility in the CaM-bound state, making it more difficult to determine the extent of domain closure in these complexes relative to those with peptides

(Prozialeck and Weiss, 1982; Craven et al., 1996; Osawa et al., 1998). Some of these antagonists, such as the antipsychotic drug trifluoperazine (TFP), the highly selective inhibitors of CaM-mediated processes W-7, J-8, calmidazolium, the arylalkylamine derivative DPD, and certain bifunctional ligands, bind CaM with affinities in the nanomolar to low micromolar range. These ligands form complexes that are often characterized by a higher degree of proximity of the two CaM domains compared with the complexes between the protein and low affinity ligands (Reid et al., 1990; Osawa et al., 1998; Osawa et al.,

1999; Trevitt et al., 2005; Kovesi et al., 2008). In contrast, an alternative mode of binding has also been reported for the high affinity antagonist Kar-2 that does not involve the hydrophobic pockets of CaM (Horvath et al., 2005).

In the present study we show that the complex between CaM and the high affinity antagonist idoxifene represents a yet different binding mode of a CaM antagonist. Idoxifene is a triphenylethylene derivative analogue of tamoxifen (Fig. 1), one of the first agents of choice for the treatment and prevention of breast cancer (Marshall, 1998; Powles, 2013). The classical

view of the mechanism of action of tamoxifen is that it competes with estradiol for binding to the estrogen receptor (ER) (Shiau et al., 1998). However, several studies indicate that tamoxifen inhibition of breast cancer growth is the result of a complex interplay involving both ER binding and CaM antagonism (Gulino et al., 1986; Cifuentes et al., 2004; Li and Sacks, 2007; Gallo et al., 2008). In addition to its therapeutic efficacy for breast cancer, tamoxifen also has antifungal activity, and inhibits the growth of various tumours by a complex mechanism that requires CaM antagonism in all cases (Cifuentes et al., 2004;

Dolan et al., 2009; Pawar et al., 2009; Byer et al., 2011). Hence, an understanding of the structural determinants of tamoxifen binding to CaM may help in the development of yet more effective therapies. To date there are few such studies: insights into the nature of the complex come mainly from molecular modelling and structure activity relationships studies (SARs) (Edwards et al., 1992; Hardcastle et al., 1995; Hardcastle et al., 1996). These studies led to the synthesis of idoxifene, a derivative of tamoxifen in which the basic dimethyl amino sidechain has been replaced by a pyrrolidine, and iodine has been placed in one





of the phenyl rings (Fig. 1). *In vitro*, idoxifene showed an inhibition potency for CaM of some 4-5 times that of tamoxifen, a
    higher toxicity towards ER positive MCF-7 human breast cancer cells, and higher *in vivo* clinical activity compared to
    tamoxifen (MacGregor and Jordan, 1998; Dowsett et al., 2000). These properties of idoxifene make it an interesting target for
    a structural study of its binding to CaM. Here we use heteronuclear multidimensional NMR and fluorescence spectroscopy to
    determine the binding affinity, stoichiometry and solution-binding mode of idoxifene to $Ca^{2+}$-CaM. A comparison with a

previously determined molecular model of the same complex and the structures of CaM-peptides and CaM-ligand complexes
    reveals an unusual binding mode that broadens the repertoire of recognition processes involving CaM.

**Figure 1: The chemical structures of tamoxifen and idoxifene. The numbering scheme for idoxifene used here is shown: phenyl
group = H7-H11; ethyl group = H12-H13; *p*-iodo-phenyl group = H15-H19; *p*-phenoxy group = H21-H25; pyrrolidine group = H28-
H31.**

## 2 Materials and Methods

### 2.1 Sample preparation

Idoxifene was a gift from the CRC Institute of Cancer Research at Sutton, Surrey UK. Unlabelled and uniformly $^{13}C/^{15}N$–
labelled recombinant mammalian CaM were prepared as described previously (Vogel et al., 1983). Purified, lyophilized,

calcium-free protein was dissolved to a concentration of 3 mM in 50 mM KCl, 5 mM $NaN_3$, 10 % $D_2O$. 3-(trimethylsilyl)
    propionic-2,2,3,3,-$d_4$ acid, sodium salt (TSP, 0.1mM) was added as a reference compound. Appropriate quantities of $CaCl_2$
    solution (6 moles per mole of CaM) were added to yield calcium-saturated protein, and checked by observing diagnostic amide
    chemical shift changes in the 1D $^1H$ NMR spectrum. The final sample volume was 450 µL. The pH of the solution was adjusted
    to 6.0 (uncorrected value) by adding microlitre quantities of 0.1 mM HCl and NaOH. CaM samples were otherwise unbuffered

because the *apo*-protein has sufficient buffering capacity at pH 6.0 and the extrinsic buffers available without non-
    exchangeable protons compete for calcium binding. The KCl concentration used here mimics conditions in which the calcium
    affinity of CaM was determined (Linse et al., 1991) and was chosen to minimise the likelihood of idoxifene precipitation at
    the end of the titration when excess ligand is present. This unbuffered system and salt concentration are in keeping with
    previous studies of CaM (50-100 mM KCl) (Finn et al., 1995; Craven et al., 1996; Trevitt et al., 2005).



Stock solutions of idoxifene were prepared at a concentration of 60 mM in CD$_3$OD. Idoxifene was added to the CaM solution up to a maximum ligand:protein molar ratio of 2.4:1. $^1$H and 2D $^1$H-$^{15}$N HSQC spectra were recorded on the protein solutions, and for each successive addition of idoxifene, in steps of 0.2 equivalents of idoxifene to CaM. A control titration with CD$_3$OD was carried out under the same conditions to correct for the small effects of CD$_3$OD on the protein chemical shifts. The pH of the protein solutions was monitored throughout the titrations, and when necessary small additions of acid or

base were made to maintain the pH at 6.0.

**2.2 Fluorescence spectroscopy**

Binding of CaM to idoxifene was monitored using changes in the intrinsic fluorescence of idoxifene at 293 K. Titrations were performed using a Cary Eclipse fluorescence spectrophotometer, with excitation at 295 nm and observation of emission at 450 nm. Idoxifene and CaM solutions were prepared in 45 mM KCl, 9 mM CaCl$_2$, pH 6.0 with 10% methanol, and CaM was added

to idoxifene in aliquots of 0.1 molar equivalents. Titrations were performed with 10 µM and 0.54 µM idoxifene, using 90 µM and 4.5 µM CaM, respectively. Concentrations were verified by quantitative 1D $^1$H NMR relative to TSP for idoxifene and by absorbance at 276 nm for CaM, using the extinction coefficient 3300 cm$^{-1}$ M$^{-1}$. The 0.54 µM idoxifene titration was fitted for $K_d$ using a 2:1 idoxifene:CaM stoichiometry, and independently fitted for both $K_d$ and stoichiometry, using in-house software.

**2.3 NMR spectroscopy**

NMR experiments were carried out at 310 K. The H$_2$O resonance was suppressed by on-resonance low power presaturation (typically applying a 10Hz field for 800 ms) during the relaxation delay, followed by a SCUBA sequence employing two composite Π pulses separated by 30 ms delay (Brown et al., 1988). The data were acquired using standard heteronuclear NMR experiments, processed using the program FELIX and deconvoluted as described previously (Craven et al., 1996). Proton, carbon and nitrogen frequencies were referenced relative to TSP, using values of 0.251449530 and 0.101329118 for $\gamma_C/\gamma_H$ and

$\gamma_N/\gamma_H$ respectively (Wishart et al., 1995).

**2.3.1 Resonance and NOE Assignment**

Resonance assignment of free CaM was based on the assignment of *Drosophila* CaM (BMRB entry 547) and it was verified using TOCSY and NOESY-HSQC experiments (Craven et al., 1996). For the idoxifene:CaM complex, the amide $^1$H-$^{15}$N correlations were followed in the titration series of HSQC spectra using 3D TOCSY-HSQC and NOESY-HSQC experiments.

The assignments of backbone and sidechain resonances of $^{13}$C,$^{15}$N-labeled CaM in the 2:1 idoxifene:CaM complex were confirmed using previously described protocols (Craven et al., 1996; Osawa et al., 1998), including 3D Long Range Carbon Correlation and 3D $^{13}$C-edited NOESY-HSQC experiments for the assignment of the ε methyl resonances of methionine. The identity of protein resonances giving intermolecular NOEs in the complex was discerned from the 3D ω$_1$-$^{13}$C-half-filtered $^{13}$C-edited NOESY-HSQC spectrum. For matching to the assignment data, tolerances of 0.03 ppm and 0.3 ppm were used for $^1$H



and $^{13}$C frequencies respectively. The 3D $^{13}$C-half filtered-$^{13}$C-edited NOESY spectrum was acquired with a mixing time of 100 ms, in line with previous solution structures of CaM-small ligand complexes (Craven et al., 1996; Osawa et al., 1998). The chemical shift values of the peak centres were converted to X-PLOR restraints using in-house software. For the idoxifene:CaM complex, 180 ligand-protein NOEs were observed, of which 110 were unambiguously assigned.

The assignment of idoxifene in the absence of protein (Fig. S1) was carried out in $D_2O$ at pH 3.0, as the ligand is insoluble

at pH 6.0. Idoxifene resonances in the complex were assigned using 2D TOCSY and 2D $^{13}$C-double-half filtered NOESY experiments. Due to spectral overlap with protein resonances, a $\omega_2$-$^{13}$C-half filtered $\omega_1$-$^{13}$C-NOESY spectrum was used to assign idoxifene resonance giving NOEs in the complex.

### 2.3.2 Lineshape Analysis

Line shapes were calculated for a simple two-state exchange model using standard equations (McConnell, 1958). The

transverse relaxation times were adjusted to match the observed line width, and the value of the off-rate was varied to optimize the agreement between the calculated and experimental data. A normalizing factor was applied to all data to correct for the constant intensity loss observed throughout the titration as a result of dilution. This was calculated by determining the mean intensity loss of a number of peaks for which no change in chemical shifts was seen on binding idoxifene and hence were unaffected by the exchange processes. In interpreting the line-shape analysis in terms of absolute stoichiometry, it is imperative

to be certain of the ligand and protein concentrations used. For the protein this was initially determined using UV absorbance and for the ligand using dry weight. The final concentration in the 2:1 complexes was then checked by comparison of peaks in 1D spectrum, which confirmed that the concentrations were correct to within 10%.

### 2.4 Structure Calculations

Restraints were classified as strong, medium or weak, and were assigned upper bounds of (i) 2.5 Å, 3.5 Å and 5.0 Å, (ii) 3.0

Å, 4.0 Å and 5.0 Å, or (iii) all as 5.0 Å, in separate calculations to accommodate the inherent uncertainty involved in converting crosspeak intensities to more precise distances when there is the possibility of conformational exchange of the ligand in its binding site. All NOE restraints were introduced using $1/r^6$ sum averaging to accommodate reorientation of aromatic rings and isopropyl groups *via* bond rotation. Structures were calculated with X-PLOR 3.1 using a molecular dynamics simulated annealing protocol for conventional protein structure determination (Brunger, 1992; Nilges, 1995). For the first 3 ps, a high

temperature (1500K) was maintained, and the weight on the core repulsion potential energy term was kept very low. This was followed by an 18 ps cooling stage in which the temperature was reduced in 50K steps, and the weight on the core repulsion term was gradually increased. Finally, the structures were subjected to 250 steps of conjugate gradient energy minimization. As a refinement stage, the temperature was increased to 1500K, and the above cycle repeated. During the high temperature stage of the protocol, a square-well NOE potential was used, with harmonic sides. During the second part of the protocol, the

X-PLOR *soft-square* potential was used, which smoothly changes the harmonic potential to a linear potential for large restraint violations. The energy constant for the harmonic potential was 5.0 kcal/mol/Å$^2$. The slope of the asymptote was 0.5 kcal/mol/Å.



The switching region between the two regimes was approximately between 0.5 and 2 Å above the upper restraint bound. The *parallhdg.pro* parameter set of X-PLOR was used, with the X-PLOR quartic *repel* potential to represent the repulsive part on the interatomic interactions. No attractive or electrostatic terms were used. The final weight on the repulsive term was 4 kcal/mol/Å$^4$.

For *ab initio* structure calculations, extended protein coordinates with random initial velocities were used as a seed, and two idoxifene molecules were placed at random within a box of side 60 Å centred on the centre of mass of CaM. For NOE-restrained docking calculations protein coordinates were taken from either the 2.4 Å resolution structure of CaM in complex with a myosin light chain kinase peptide (pdb entry 1cdl (Meador et al., 1992)), or the tr2c domain of the 1.7 Å resolution X-ray structure of mammalian CaM (pdb entry 1cll (Chattopadhyaya et al., 1992)). Two sets of starting positions of the idoxifene molecules were investigated – molecules placed at random in a box, and molecules manually docked into the hydrophobic pocket – to exclude any bias away from occupancy of the hydrophobic pocket through restrictions in the sampling of the relative positions of molecules. For the former, a fresh idoxifene starting conformation for each calculation was created by first transforming the coordinates in an extended conformation by a random rigid body rotation. The centre of mass was then placed at random within a box of side 60 Å centred on the centre of mass of the protein molecule. The idoxifene molecule was treated as flexible, subject to restraints of covalent geometry and van der Waals contacts. The sidechains involved in intermolecular NOEs were either fixed with the remainder of the protein, or allowed the same internal flexibility as the idoxifene molecules, and the two calculations were compared. For the manual docking of idoxifene molecules into the hydrophobic pocket of the protein, 50 structures were generated in which different hydrophobic parts of the idoxifene molecule were placed deep inside the hydrophobic pocket of the domain, and the molecules were subjected to 250 steps of conjugate gradient energy minimization to remove steric clashes. In each case part of the idoxifene molecule remained within the hydrophobic pocket. These structures were then used as starting coordinates for the structure calculation protocols described above. For both sets of starting positions, the final distribution of structures was indistinguishable. When more than one idoxifene molecule per domain of CaM was included in the calculation, the core repulsion terms between idoxifene molecules was set to zero whereas those within each idoxifene and between each idoxifene and the protein were increased during the course of the calculation as described above. This meant that there was no energy penalty to atoms from more than one idoxifene molecule occupying the same space. An NOE restraint was satisfied by the proximity to the protein atoms of atoms from any individual idoxifene molecule.

## 3 Results

### 3.1 Titration of CaM with idoxifene

The binding stoichiometry and affinity between idoxifene and CaM were measured using changes in idoxifene fluorescence at 450 nm upon addition of increasing amounts of $Ca^{2+}$-CaM. Using 10 μM idoxifene, the response on CaM addition was linear up to a stoichiometry of 2:1 idoxifene:CaM (Fig. 2A). Using 540 nM idoxifene (Fig. 2B), the data fit to a $K_d$ of 340 +/- 30 nM





using a stoichiometry of 2:1 idoxifene:CaM. When both $K_d$ and stoichiometry were allow to vary, the data best fit to a $K_d$ of
180 +/- 50 nM and a stoichiometry of 1.7:1 idoxifene:CaM but there is co-variance between these parameters in the range that
includes a 2:1 stoichiometry and a $K_d$ of 340 nM. A chi$^2$ analysis of fits indicated a clear minimum at $K_d$ = 300 nM. The binding
of idoxifene to CaM was monitored independently using 1D $^1$H NMR, and the perturbations of low field amide $^1$H resonances
during the addition of up to 2.4 equivalents of idoxifene are shown in Fig. 2C. Where the chemical shift changes induced by
complex formation are much greater than 0.05 ppm, two sets of resonances are detected during the titration (e.g. I27, I100,
G134 in Fig. 2C). Thus, for these resonances, the dissociation rate of the complex is predominantly in the slow exchange
regime on the NMR timescale. The resonances corresponding to free CaM disappeared when 2 equivalents of ligand were
added, in line with the stoichiometry of 2:1 reported by the fluorescence measurements using 10 μM idoxifene. Some
resonances experienced a reduction of peak height at intermediate ligand concentrations, and re-sharpened when two
equivalents of idoxifene were added (e.g. D64, and N137 in Fig. 2C). The extent of line-broadening allowed an estimate for
the off-rate of $30 \pm 10$ s$^{-1}$ to be obtained, which, in combination with the measured $K_d$ value, indicates that the complex forms
with a diffusion-controlled on-rate of *ca*. 1 x 10$^8$ M$^{-1}$s$^{-1}$. D64 and N137 occupy equivalent positions in the CaM structure,
being located in position 9 of Ca$^{2+}$ binding loops II and IV, and contribute to the 3 residue β-strands in each domain. The
resonances of T28 and S101, the equivalent residues in loops I and III, show slow exchange behavior as do all of the other
residues in the short β-sheets of CaM (e.g. I27 and I100 in Fig. 2C). The anomalous behavior of D64 and N137 may result
from a sensitivity of these residues to the occupation of the other globular domain of CaM by idoxifene.

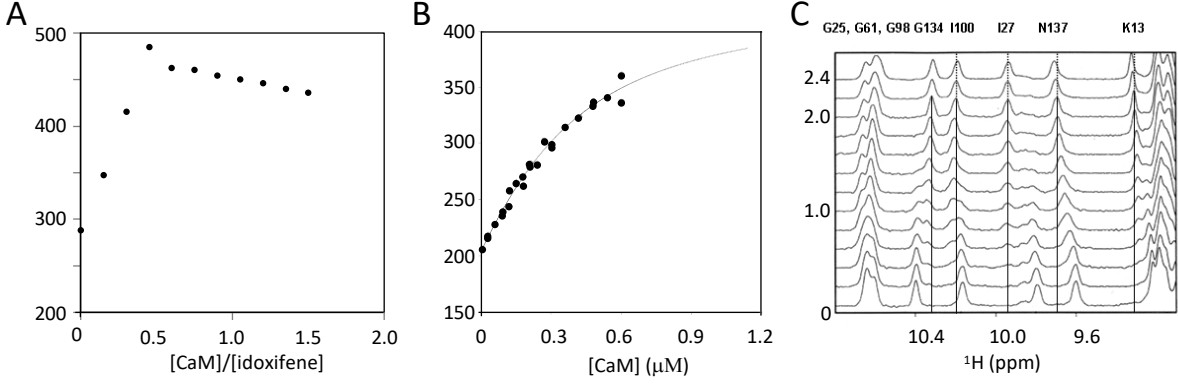

**Figure 2: Titration of idoxifene with calmodulin. (A)** The changes in fluorescence intensity of 10 μM idoxifene at 450 nm on addition
of CaM up to 15 μM. Saturation is reached at a concentration ratio of CaM to Idoxifene of 0.5, indicating the presence of two binding
sites with sub-micromolar binding affinity. **(B)** Repeat of (A) but with 0.54 μM idoxifene and increasing CaM concentrations from
0.02 μM up to 0.6 μM. The data were fitted to a binding isotherm with a 2:1 idoxifene:CaM binding stoichiometry. **(C)** A region of
the 1D $^1$H spectra acquired during the titration of idoxifene into 3 mM CaM. Each spectrum corresponds to the addition of 0.2
equivalents of idoxifene into CaM, up to 2.4 equivalents (left scale). Solid lines highlight the completion of the slow exchange event
for the assigned resonances at 2:1 idoxifene:CaM. Dotted lines are drawn for resonances undergoing shift changes beyond the end
point of the titration. G134, I100, I27 are representative of the slow exchange regime observed for CaM resonances during the
titration, while N137 is representative of intermediate exchange character.



## 3.2 Chemical Shift Changes

The NMR resonances of free idoxifene were assigned on the basis of characteristic chemical shifts of model compounds (2-pyrrolidinoethanol, iodo-benzene, methoxy-benzene), NOEs and correlations observed in a 2D $^1$H TOCSY spectrum (Table 1 and Fig. S1). The backbone and sidechain resonances of $^{13}$C,$^{15}$N-labeled CaM in the 2:1 idoxifene:CaM complex were assigned

using previously described protocols (Craven et al., 1996; Osawa et al., 1998). The acquisition of intra and inter residue carbonyl shifts was essential in order to distinguish residues with degenerate Cα shifts, such as D50 and D122. More severe overlap present in the aromatic region of the spectra of the complex prevented the assignment of the ζ resonances of the phenylalanine residues. The assignment of the ε methyl resonances of methionine residues was achieved using a combination of 3D Long Range Carbon Correlation (LRCC) and 3D $^{13}$C-edited NOESY-HSQC experiments (seven of the nine were

assigned). Some resonances from residues L69-K77 were attenuated, particularly their Cα resonances, indicative of conformational exchange in the flexible tether region between domains in the complex.

**Table 1. $^1$H chemical shifts of idoxifene, free in solution and bound to CaM.**

| Resonance | Free* | Bound† | |
|---|---|---|---|
| $^1$H | δ | δ | Δδ |
| 16/18 | 7.52 | 7.42 | -0.10 |
| 15/19 | 6.86 | 6.84 | -0.02 |
| 21/25 | 6.69 | 6.71 | +0.02 |
| 22/24 | 6.47 | 6.55 | +0.08 |
| 7/11 | 6.99 | 6.98 | -0.01 |
| 8, 9, 10‡ | 7.06 | 7.08 | +0.02 |
| 12 | 2.23 | 2.31 | +0.08 |
| 13 | 0.73 | 0.78 | +0.05 |
| 26 | 3.96 | 4.10 | +0.14 |
| 27 | 3.43 | 3.50 | +0.07 |
| 28/31 | 3.39/3.06 | 3.30 | +0.07 |
| 29/30 | 1.96 | 1.99 | +0.03 |

* D$_2$O, pH 3.0. † D$_2$O, pH 6.0. ‡ Unresolved resonances.

A summary of the backbone chemical shift changes observed on binding of idoxifene to CaM is shown in Fig. 3 as a weighted average (WA) of the changes of all five backbone resonances, obtained using the equation:





$$|WA| \cong \sum |\delta_{CaM:drug} - \delta_{CaM}|/|\Delta_{max}| \qquad (1)$$

where the summation extends over the backbone atoms, $\delta_{CaM:drug}$ and $\delta_{CaM}$ are the chemical shifts observed in the complex and

in free CaM, and $\Delta_{max}$ is the largest chemical shift change observed for each type of nucleus. The chemical shift changes of

each of the five backbone atoms are shown in Fig. S2.

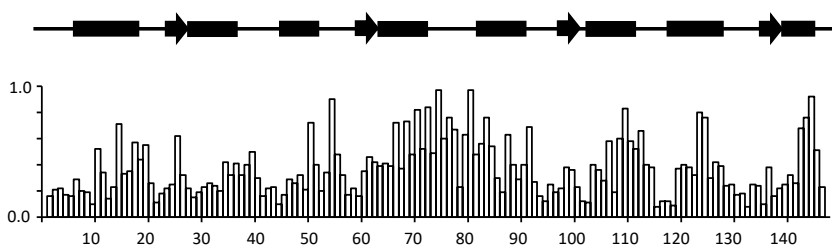

**Figure 3: Chemical shift changes for backbone CaM resonances upon binding idoxifene. Normalised, weighted average chemical shift changes are plotted against the primary sequence of CaM. Secondary structure elements are indicated above the histogram as solid boxes for α-helices, solid arrows for β-strands and thin lines for unstructured regions. The flexible tether (E67-V85)**
**experiences the largest chemical shifts changes upon complex formation.**

Substantial chemical shift changes are seen in many contiguous stretches of the backbone, such as E67-I85 and F141-T146

and are mapped onto the open structure of CaM in Fig. 4A. In contrast, some residues such as V55 display large shift changes

whereas surrounding residues are hardly perturbed. Residues E67-I85 include the flexible tether between the two domains and

the chemical shift changes here are closely comparable with those in the same region observed on formation of the CaM:M13

complex (Ikura et al., 1991), where CaM wraps its two domains around a helical peptide in a compact, globular structure.

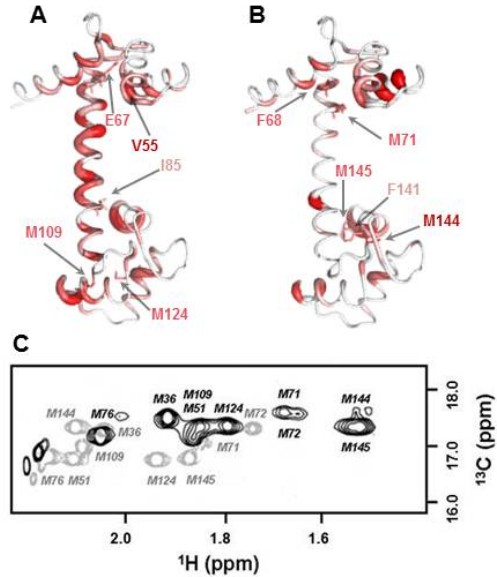



**Figure 4:** Comparison of backbone and sidechain chemical shift changes of CaM resonances upon binding idoxifene. (A) and (B), heat map representations of CaM chemical shift changes upon idoxifene binding, shown for clarity using the coordinates of the unliganded CaM X-ray structure, 1cll. The hydrophobic pocket of the N-terminus at the top and the rear of the C-terminal hydrophobic pocket at the bottom. In (A) residues are coloured according to the weighted average (WA) backbone shift changes observed in the idoxifene:CaM complex, with gradation from white (WA shift = 0.0 ppm) to red (WA shift > 0.7 ppm) and the broadest ribbon indicating the maximum shift changes. The locations of M109 and M124 by the hydrophobic pocket of the C-domain are indicated. Residues I85 and E67 are indicated, as they define the central portion of the flexible tether joining the two domains. The position of V55, which undergoes a large shift change as in the M13:CaM complex, is also indicated. In (B) residues are coloured according to the largest sidechain $^1$H shifts observed upon idoxifene binding (shifts > 0.50 ppm), with colour gradation and ribbon width depicted as in (A). The locations of key residues such as three of the nine methionine in the vicinity of the hydrophobic pockets and F68, F141 are indicated. (C) a 2D $^{13}$C CT-HSQC spectra showing the ε methyl methionine signals of free CaM (grey) and idoxifene bound CaM (black).

The positions of the sidechain chemical shift changes observed on binding of idoxifene to CaM are shown in Fig. 4B. The majority of protein sidechain resonances move by less than 0.05 ppm for $^1$H and 0.3 ppm for $^{13}$C. The larger sidechain shift changes are confined to residues around the hydrophobic pockets of both CaM domains, with the largest changes observed for methionine residues, in particular the β and ε resonances of M71 and M144 (Fig. 4C), which occupy equivalent positions near the C-terminus of each domain, on the rim of their hydrophobic pockets. Also of note are the chemical shift changes in phenylalanine ring resonances. Only F68 and F141 undergo large shifts upon idoxifene binding, indicating that the structure of the individual domains is not significantly perturbed in the complex. The aromatic rings of F68 and F141 directly contact the ε methyl groups of M71 and M144, and the combined chemical shift perturbations of these residues indicate a local disturbance in this region upon idoxifene binding. Overall, it is clear that while sidechain shift changes are localized around the hydrophobic pockets of both domains, backbone shift changes extend to the tether and the rear of the domains.

### 3.3 Structure Determination

Intramolecular NOEs within CaM, and intermolecular NOEs between idoxifene and CaM were quantified using $^{13}$C- and $^{15}$N-edited NOESY-HSQC spectra. Intermolecular NOEs were distinguished from intra-protein, and from intra-ligand NOEs, using isotope filtering. A portion of a 2D $^{13}$C-$^{15}$N-double-filtered NOESY spectrum of a 2:1 idoxifene:CaM solution is shown in Fig. 5. Most of the intermolecular NOEs arising from the phenyl, *p*-iodo-phenyl, *p*-phenoxy and ethyl group of the idoxifene (Fig. 1) are to protein sidechain resonances in the vicinity of the two hydrophobic pockets exposed on calcium binding; none are observed to the opposite faces of the CaM domains. However, it is immediately striking that the phenyl, *p*-iodo-phenyl, *p*-phenoxy and ethyl groups mostly have substantial NOEs to the same resonances on the protein, despite those resonances belonging to nuclei that are distributed widely across each domain (Fig. S3). Such an NOE distribution can be associated with elevated levels of spin-diffusion at extended mixing times, but the mixing time employed here (100 ms) is the same as or shorter than that used in the structure determination of other CaM-ligand complexes (Craven et al., 1996, Osawa et al., 1998; Osawa et al., 1999), where spin-diffusion was not significant, and there is no significant difference in rotational correlation times between these complexes. Moreover, spin-diffusion effects are not dominant between nuclei within the ligand when it is bound in the complex. Hence, spin-diffusion alone cannot account for the unusual similarity in intensities of the intermolecular NOEs involving the phenyl, *p*-iodo-phenyl, *p*-phenoxy and ethyl groups.



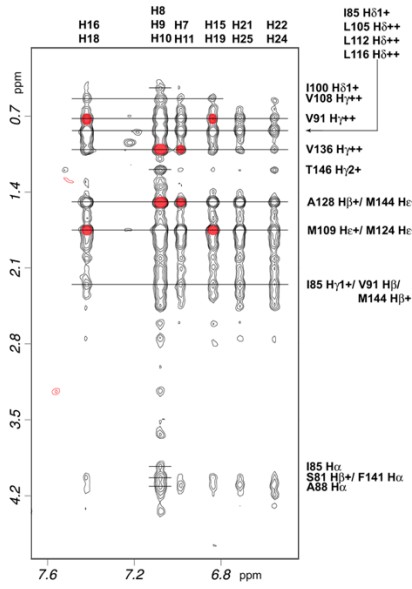

**Figure 5: Intermolecular NOEs between idoxifene and CaM.** A region of the filtered 2D NOESY spectrum of the 2:1 idoxifene:CaM complex, processed to exclude the $^{13}$C/$^{15}$N bound resonances in the $\omega_2$ dimension and to include only the $^{13}$C bound resonances in $\omega_1$ dimension. The ligand resonances corresponding to each strip are marked on the top edge as follows; *p*-iodo-phenyl ring (H16/H18), phenyl ring (H8/H9/H10), phenyl ring (H7/H11), *p*-iodo-phenyl ring (H15/19) and *p*-phenoxy ring (H21/25 and H22/24). Assignments of the resonances from tr2c are shown on the right. A red circle over the crosspeak indicates examples of NOEs that are violated in the structure calculation corresponding to the data in Fig. S3.

Initially, standard NOE-restrained structure calculations were performed including intra-protein, intra-ligand and intermolecular NOEs, classified as strong, medium or weak according to intensity (assigned upper bounds of 2.5 Å, 3.5 Å and 5.0 Å). The resulting structures did not converge to a well-defined ensemble and only high-energy structures with widespread NOE violations resulted. Since the conformations of the two individual domains of mammalian CaM are almost invariant across all deposited X-ray structures and the intra-protein NOE distribution within the domains of the 2:1 idoxifene:CaM complex reflected closely that of free CaM, focus was shifted to a restrained docking strategy. This allowed the NOE violation energies in the restrained dynamics to be isolated to the intermolecular NOEs rather than be spread widely across the CaM domains.

In contrast to the invariance of individual domain structures within CaM complexes, the relative position of the two domains varies considerably between deposited structures. NOE-restrained structure calculations where the individual domain structures were fixed but the flexible tether between the two domains was allowed conformational freedom led to closed structures of the idoxifene:CaM complex. However, there was little convergence in the relative position of the two domains and widespread intermolecular NOE violations remained. Consequently, NOE-restrained docking to a closed structure that well represented the chemical shift changes in the protein backbone (*c.f.* Fig. 3 and Fig. S2), that of the CaM:M13 (myosin light chain kinase peptide) complex, was investigated. With the intermolecular NOEs calibrated directly according to their



intensities, the docked structures distributed the idoxifene molecules widely across the hydrophobic surfaces of the two domains of CaM but 64 of the 180 NOEs remained violated in the lowest energy structure. When the intermolecular NOEs were all assigned upper bounds of 5.0 Å and the contacting sidechains within the individual domains of CaM were allowed conformational flexibility, the lowest energy structures converged, whereby the idoxifene molecules occupied one of two sites

on CaM in the vicinity of the hydrophobic pocket on each domain (Fig. 6A-B), and the NOE violations in the lowest energy structures dropped to an average of 5. However, while the docked idoxifene molecules delineated two binding sites in CaM, the orientation of the individual idoxifene molecules was very variable between structures.

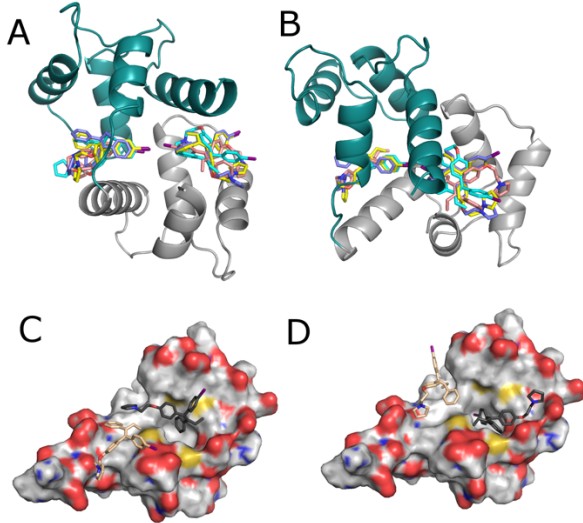

**Figure 6: Representative 2:1 idoxifene:CaM complex structures.** (A) and (B) are orthogonal views of the 2:1 idoxifene:CaM complex calculated using NOE-restrained docking of two molecules of idoxifene to CaM coordinates based on the CaM:M13 complex (pdb entry 1cdl). In this calculation, the upper bounds of all observed intermolecular NOEs were set to 5 Å. CaM domains tr1c and tr2c are coloured green and grey, respectively, and the idoxifene molecules are coloured in pairs (blue, cyan, yellow or beige) derived from single calculations. View B is orientated to coincide with parts C and D. (C) and (D) show the positions of four molecules of idoxifene derived from a single NOE-restrained docking calculation to the tr2c domain of CaM (from pdb entry 1cll), illustrating one arrangement of 4 conformations that satisfy the NOE restraints. The four idoxifene molecules are shown in pairs for clarity only. The two darker coloured idoxifene molecules have either the phenyl group (C) or the *p*-iodo-phenyl group (D) by the pocket in the hydrophobic surface of the domain. The lighter coloured molecules also satisfy NOEs that are within reach of the ligand binding site associated with the tr1c domain. The surface representation of tr2c is coloured as follows: carbon (*grey*), nitrogen (*blue*), oxygen (*red*) and sulphur (*yellow*).

In order to establish whether the variability of orientation of the idoxifene ligands in their sites could be reduced, a gradation of upper bound limits according to intensities of the intermolecular NOEs was re-introduced. Increased deviation from a uniform upper bound of 5.0 Å led to increased restraint violations and, in general, pulled the idoxifene molecules towards the centre of mass of CaM, but without any discernible decrease in the variability of the orientation of individual idoxifene molecules. An alternative approach was thus investigated where the protein in the docking protocol was simplified to one

domain - the tr2c domain. This approach removes NOE violation energies that inadvertently pull an idoxifene molecule





towards the binding site of the other domain. With the intermolecular NOEs assigned upper bounds of 2.5 Å, 3.5 Å and 5.0 Å, a minimum of 26 of 82 NOEs were violated.

The high proportion of intermolecular NOE violations, coupled with the substantial number of NOEs between distant parts of the ligand and the same resonances on the protein (Fig. 5), points to a complex where the ligand is undergoing
conformational exchange while bound to the protein. In order to simulate this in the NOE-restrained docking, the number of idoxifene molecules included was increased until all of the NOEs assigned upper bounds of 2.5 Å, 3.5 Å and 5.0 Å were satisfied. No interaction potentials were included between idoxifene molecules in order that multiple binding conformations could be satisfied simultaneously in the calculations, where ligands occupied the same space. With the introduction of four idoxifene molecules, all of the NOE restraints could be satisfied. A representative structure with no NOE violations is shown
in Fig. 6C-D. The NOEs are satisfied by idoxifene placing either the *p*-iodo-phenyl (Fig. 6C) or the phenyl group (Fig. 6D) by the pocket in the hydrophobic surface of the domain and orienting the *p*-phenoxy and ethyl group alternately towards or away from the position that would be occupied by the tr1c domain (c.f. Fig. 6C-D and Fig. 6B). The two other idoxifene molecules also satisfy NOEs that are within reach of the ligand-binding site associated with the tr1c domain (left side of Fig. 6B). When this calculation is repeated with full-length CaM (in the closed form derived from CaM:M13 complex), the NOE restraints
were satisfied using four idoxifene molecules per domain and the positions of the idoxifene molecules in the full length complex followed the pattern observed in the binding to tr2c, where either the *p*-iodo-phenyl or the phenyl group occupy the mouth of the hydrophobic pocket and the *p*-phenoxy and ethyl group lay along the hydrophobic surface in one of two directions (Fig. 6).

## 4 Discussion

There is now a considerable body of data on CaM-ligand interactions, including solution and solid-state studies on the interactions with both peptides and small molecules (Meador et al., 1992; Ikura et al., 1992; Vandonselaar et al., 1994; Craven et al., 1996; Osawa et al., 1998; Harmat et al., 2000; Horvath et al., 2005; Kovesi et al., 2008). CaM-peptide interactions are normally satisfactorily described by a single set of coordinates for the ligand, representing an average structure, and this is reflected in the good agreement between the structure of these complexes determined using X-ray and NMR spectroscopy
methods (Ikura et al., 1991; Meador et al., 1992; Ikura et al., 1992; Maximciuc et al., 2006). However, this approach is less appropriate in the case of many CaM-small ligand interactions. For example, two DPD:CaM complex structures show substantial differences in the orientation of the ligand (Fig. S4B) and extent of CaM domain closure, while computation of the same complex suggests that CaM adopts a more globular structure in solution relative to the conformations observed in the X-ray structures (Harmat et al., 2000; Kovesi et al., 2008). Other anomalies also often appear in CaM-small ligand complexes.
For example, while X-ray structures of TFP:CaM complexes have been solved with one, two and four TFPs bound to CaM, NMR measurements and computation show conclusively that the principal binding mode in solution involves two TFP molecules bound with indistinguishable affinity, one to the equivalent site in each domain (Vandonselaar et al., 1994; Craven





et al., 1996). Both the above scenarios illustrate that CaM-small ligand complexes normally require a description including a combination of stoichiometry, relative affinity, exchange dynamics and structure.

### 4.1 Binding affinity and stoichiometry

The principal binding mode of idoxifene to CaM has a stoichiometry of 2:1 (according to both fluorescence and NMR measurements) and a dissociation constant of ~300 nM, resulting from a diffusion-controlled on-rate of ~$10^8$ M$^{-1}$s$^{-1}$ and a ligand dissociation rate of ~30 s$^{-1}$. This dissociation constant indicates that idoxifene has at least an order of magnitude stronger affinity for CaM than that observed for most other antagonists, with the exception of calmidazolium ($K_d$ ~1-10 nM), DPD ($K_d$ ~18 nM), and other bifunctional ligands (Reid et al., 1990; Harmat et al., 2000; Trevitt et al., 2005). The $K_d$ value determined here is lower than the previously reported IC$_{50}$ value for idoxifene, 1.5 μM (Hardcastle et al., 1996), which was derived indirectly from the inhibition of calmodulin-dependent cyclic AMP phosphodiesterase, and under experimental conditions different from here. On addition of more than two equivalents of idoxifene further small chemical shift changes are observed for some of the CaM resonances (e.g. K13 and N137 in Fig. 2) in the fast exchange regime on the NMR timescale. Such behaviour has been observed also for other small ligands binding to CaM, such as W-7, J-8 and TFP (Craven et al., 1996; Osawa et al., 1998) and is attributed to secondary, weaker binding phenomena.

### 4.2 Domain closure

The evidence points to CaM closing to a compact, globular structure on binding idoxifene. The observed distribution and magnitude of backbone chemical shift changes in the 2:1 idoxifene:CaM complex are closely similar to those on formation of the CaM:M13 complex, which forms a tightly closed, compact conformation (Ikura et al., 1991; Barbato et al., 1992). In CaM complexes in which the ligand binds to only one of the two domains and the complex does not close, the chemical shift changes of the vast majority of the amide protons in the central linker are very small, usually less than 0.1 ppm (Elshorst et al., 1999). No direct NOEs were observed between the two domains of CaM in the 2:1 idoxifene:CaM complex, but this was also the case for the W-7:CaM complex, which forms a tightly closed conformation similar to that of the CaM:M13 complex (Osawa et al., 1998; Osawa et al., 1999). Indeed, the direction and magnitude of the backbone protons chemical shift changes for residues in the flexible linker (e.g. E82 and D80) in the W-7:CaM complex are similar to those observed on formation of the 2:1 idoxifene:CaM complex (c.f. Fig. S2 and Fig. 6 in Osawa et al., 1999). Moreover, the backbone chemical shift changes within the central tether in the idoxifene:CaM complex cannot be attributed solely to proximity to idoxifene - the intermolecular NOEs and the largest sidechain chemical shift changes are not located in the tether (Fig. 4A-B). In addition, backbone chemical shifts changes for T110-L112 are indicative of the formation of α-helical structure, while those of M144-T146 are indicative of a loss of α-helical structure, as reported for the CaM:M13 complex (Ikura et al., 1991). Finally, in the NOE-restrained docking calculations, when the residues in the flexible tether region were allowed conformational freedom, the lowest energy structures were always closed.





### 4.3 Ligand distribution

The distribution of NOEs between CaM and idoxifene resonances reflects closely the distribution of the largest chemical shift changes of protein sidechain resonances observed upon idoxifene binding. The contributing sidechains are located around the hydrophobic pockets of both CaM domains and no large chemical shift changes are observed for sidechains in the tether or the rear of the domains (Fig. 4B). The strongest intermolecular NOEs were to the ε methyl groups of methionine residues and these resonances undergo the largest chemical shift changes (Fig. 4C). Substantial upfield chemical shift changes of these

methionine resonances has been reported for numerous peptide and ligand complexes of CaM, and attributed to ring current effects from the aromatic groups of the bound ligand. It is well established that the high abundance of methionines is essential in the biological function of CaM, the sidechains of which can provide the conformational variability that enables the binding of a wide range of ligands (Osawa et al., 1998; Elshorst et al., 1999).

The lack of substantial chemical shift perturbations in the phenylalanine rings in the vicinity of the hydrophobic pockets
and the low number of NOEs observed between phenylalanine residues and idoxifene indicate that the ligand does not occupy the hydrophobic pockets fully (Fig. 4B). Only F68 and F141 significantly change chemical shift on idoxifene binding and these changes reflect the loss of α-helical structure in the M71-A73 and M144-T146 regions that is typically seen on peptide binding (Ikura et al., 1991). A likely driver for this loss of α-helix is that the unwinding allows the methyl group of, for example, T146 to join the hydrophobic surface that interacts with the ligand. This interpretation is supported by the observation

of strong NOEs between the hydrophobic groups of idoxifene and T146γ, and the chemical shift changes for F141 and M144, which contact each other in the tr2c domain (and F68 and M71 in the equivalent positions in the tr1c domain). Strong NOEs and large chemical shift changes were observed for other hydrophobic residues located over the surface around the hydrophobic pockets, such as I85 and A128 (Figs. 3 and 4).

Overall, the distribution of intermolecular NOEs in the 2:1 idoxifene:CaM complex is unusual (Fig. 5) in that each of the
substituents around the central double bond of idoxifene (Fig. 1) appears to be close to the same atoms of CaM, despite the size and relative rigidity of the propeller structure in this part of the ligand. Moreover, the atoms on CaM that are close to each of these substituents of idoxifene are widely distributed across the hydrophobic surfaces of the domains (Fig. S3). The unusual distribution of NOEs is not the result of spin diffusion and therefore must reflect some positional heterogeneity within the complex. As free and bound CaM are in the slow exchange regime on the NMR timescale, and only a single set of resonances

is observed for the 2:1 idoxifene:CaM complex, idoxifene must exchange between these positions while bound to CaM. This interpretation is supported by the failure of standard structure determination methods to converge to a single conformation for the complex, and by NOE-restrained docking calculations being unable to satisfy the NOEs when only one idoxifene molecule per CaM domain was present in the calculation. When the upper bounds of the NOE restraints were relaxed, the calculated structures delineated the two binding sites in the proteins, which matched well with the chemical shift perturbations observed

on idoxifene binding. However, the orientation of the idoxifene molecules were not defined by these calculations, and the introduction of less conservative upper bounds required multiple orientations of idoxifene to be satisfied.



Hence, the relatively even distribution of intermolecular NOEs (Fig. 5) is only readily explicable in terms of multiple binding orientations for the two idoxifene molecules and establishes that the complex cannot be described by a single structure. In the NOE restrained docking, a minimum of four orientations of idoxifene per CaM domain were required to satisfy the

NOEs. The positions and orientations of the four idoxifene molecules still varied between calculations but can be described approximately by a combination of four conformers (Fig. 6): two where the *p*-iodo-phenyl group occupies the mouth of the hydrophobic pocket and the *p*-phenoxy group lies in the direction either of helix 1 or of helix 7; two where the phenyl group occupies the mouth of the hydrophobic pocket and the *p*-phenoxy group has one or other of the above orientations. More precise positioning of idoxifene molecules is not readily obtained from the NOE-restrained docking calculations for several

reasons. Firstly, the relative position and orientation of the tr1c and tr2c domain may differ slightly from those in the CaM:M13 complex, and this is not readily determinable without multiple NOEs between the domains. Secondly, the conversion of NOE crosspeak intensities to distances is not unambiguous, as an accurate measurement of the relative populations of the conformers is not independently available. Thirdly, although four idoxifene molecules per domain satisfy the NOE restraints, this is a minimum number of conformers rather than a uniquely determined number.

## 4.4 Differences with previous models

The positionally dynamic mode of binding of idoxifene to CaM contrasts with the common view that small ligands bind to proteins in a single orientation when the binding affinity is sub-micromolar. However, this paradigm has been challenged by a number of studies (Chattopadhyaya et al., 1992; Carroll et al., 2011; Hughes et al., 2012) that show dynamic binding modes not unrelated to the behavior of idoxifene when bound to CaM. There are two main differences between the behavior of

idoxifene and previously determined CaM:small ligands complexes: the requirement for a broad ensemble rather than a well-positioned bound ligand and the degree of occupancy of the hydrophobic pockets. The requirement for multiple orientations of idoxifene to describe the complex was, however, proposed previously on the basis of molecular modeling (Edwards et al., 1992). Although there are some differences in the conformations and orientations of the bound idoxifene molecules, in both this and an earlier computational model, the aromatic groups of the ligand are not located deep in the hydrophobic pockets, in

contrast to the behavior of the W-7, J-8 and DPD:CaM complexes (Fig. S4). Furthermore, based on the results of the molecular modeling and subsequent SARs of idoxifene analogues it was suggested that the binding of idoxifene to CaM would produce a compact globular protein structure similar to that observed in the peptide and TFP complexes (Hardcastle et al., 1996).

## 4.5 Role of iodine

In the molecular modeling study, the increased CaM antagonism of idoxifene relative to tamoxifen was attributed to the

presence of a hydrophobic group such as iodine (Edwards et al., 1992). However, in subsequent SAR studies it was found that substitution of the iodine with a more hydrophobic group such as a butyl chain did not improve CaM antagonism (Hardcastle et al., 1995). Similarly, J-8, an analogue of W-7 with iodine in place of chlorine, was also found to be the most potent CaM antagonist in a series of derivatives bearing different halogens in the naphthalene rings (MacNeil et al., 1988). These data and





the observation that in the J-8:CaM complex the iodine is placed inside the hydrophobic pockets of the CaM domains, led to

the conclusion that the affinity for CaM correlates with the Van der Waals radius of the halogen (Craven et al., 1996). However, in the idoxifene:CaM complex, the iodine is not positioned deeply inside the hydrophobic pockets (Fig. 6C-D), but contacts the sulphur atoms and methyl groups of M109, M144 and the oxygen of E127. Similar interactions are observed between the chlorine in the ligand Kar-2 and M109 and E114 in the X-ray structure of the Kar2:CaM complex (Horvath et al., 2005). In addition Kar-2 shows a mode of binding similar to that of idoxifene: both ligands bind to the hydrophobic surface of the

domains but not to residues located deep in the hydrophobic pockets. This mode of binding is also observed for CaM in complex with a peptide derived from the myristoylated alanine-rich C kinase substrate (MARCKS), and with a peptide derived from the HIV-1 matrix protein p17 (Yamauchi et al., 2003; Izumi et al., 2008). The X-ray structure of the CaM:MARCKS complex shows that the terminal lysine sidechains of the peptide are disordered. The presence of multiple conformers is suggested to keep the peptide flexible to maximize contacts with the acidic residues located over the surface of the two

domains.

### 4.6 Role of charge

Electrostatic interactions have previously been proposed to be important in the binding of small ligands to CaM; for example, W-7, J-8, DPD, TFP and tamoxifen (Edwards et al., 1992; Vandonselaar et al., 1994; Craven et al., 1996; Osawa et al., 1998; Harmat et al., 2000) all contain a flexible chain connected to a basic nitrogen (Figs. 1 and 6). A similar picture emerges in the

idoxifene:CaM complex. The pyrrolidine ring appears to occupy multiple areas of the protein surface located around the pockets where it can contact the glutamic acid residues; intermolecular NOEs from the pyrrolidine ring to the glutamate sidechains of both domains indicates their close proximity. The two X-ray structures of CaM complexed with DPD, which has a propeller structure similar to that of idoxifene, also show multiple orientations of the analogous basic sidechain (Fig. S4B). In some idoxifene:CaM conformers, the pyrrolidine ring is oriented toward the flexible tether that connects the two domains,

bringing the pyrrolidine nitrogen close to E84 and E87. A similar orientation of the basic chain is observed in the solution structure of the W-7:CaM complex (Fig. S4C) and also proposed in the computational model of the idoxifene:CaM complex (Edwards et al., 1992).

### 5 Conclusion

The observation that four idoxifene molecules are sufficient to satisfy the NOE restraints in the closed form of full-length

CaM, argues strongly that the 2:1 complex is an ensemble where each idoxifene molecule is predominantly in the vicinity of one of the two hydrophobic patches, fluctuating between a conformational distribution. The CaM molecule is predominantly in the form where the N and C-terminal domains are in close proximity, meaning that the idoxifene molecules are able to contact both domains simultaneously. In addition, these results show that the substantial occupation of the hydrophobic pocket observed with TFP, J-8 W-7 and DPD does not appear to be an essential component of high affinity binding. This is further



supported by the observation that there are other CaM complexes with high affinity ligands that do not bind into the hydrophobic pockets. It also explains the results of many SARs on idoxifene: synthetic modification of the aromatic rings has not led to substantial improvement in CaM antagonism. The model presented in here opens up opportunities to design substantially higher affinity antagonists of CaM activity. In addition to extending the repertoire of CaM antagonism, the dynamic mode of binding adds to the growing number of similar binding modes reported recently for small molecule ligands.


**Author contribution**: LM, CRT and BW performed most of the experimental work. AMH, ST and LLPH provided expert support in NMR methodology and data analysis. LM, BW, CAH and JPW prepared the manuscript with contributions from all co-authors. The project was initiated by CAH and JPW.

**Competing interests**: The authors declare that they have no conflict of interest.

**Acknowledgements**: We thank Stephen Neidle for providing the idoxifene, Jeremy Craven and Svetlana Sedelnikova for expert technical assistance, the BBSRC, GSK and the Wellcome Trust for financial support.

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
