# Peer review of "High Affinity Tamoxifen Analogues Retain Extensive Positional Disorder when Bound to Calmodulin"

_Magnetic Resonance, 2021_

## Author Response (AR1)

**Walter Chazin and Randy Perera**

*(1) Comments*

This manuscript presents an investigation of the interaction of calmodulin with the small molecule Tamoxifen using solution NMR. Interestingly, Tamoxifen was found to bind with high affinity without occupying specific hydrophobic pockets. This was clearly evident from the inability to satisfy inter-molecular NOEs by a single structure of the complex. The experimental approach is sound and the experiments are well designed, including several well thought out controls. This includes an important titration carried out to control for chemical shift perturbations that arise from $CD_3OD$. The data are properly interpreted. The manuscript is well written and clear. The manuscript is suitable for publication requiring only a few typographical/grammar adjustments.

1. Line 184 – "were allowed"
2. Line 266 – Remove "from:
3. Line 395 – "have been"
4. Line 420 – "was"

This review was performed primarily by postdoctoral fellow Randika (Randy) Perera, Ph.D.

*(2) Author Response*

Thank you both for reviewing our manuscript. We will make the changes that you noted.

*(3) Author Changes*

Three changes have been made, one (Line 266) was redundant owing to changes below.

**Anonymous Referee #2**

*(1) Comments*

In the current manuscript, Milanesi et al. characterize the binding mode of the sub-microM binding drug idoxifene to calcium-loaded calmodulin. Interestingly, the relatively high affinity of the 2:1 complex does not preclude extensive mobility of the bound ligands. Since most titrating calmodulin resonances show slow exchange behaviour, the KD was determined using fluorescence of the ligand, whereas the localisation of the binding sites was established by NMR, based on CSPs and NOEs. Critically, variable orientations of idoxifene molecules within each of the two binding sites was implied by NMR-restrained docking to satisfy all observed intermolecular NOEs. While mostly well-established NMR method have been employed here, the obtained experimental data convincingly supports the conclusions drawn about an

unusual ligand-protein binding mode, which has significant implications for calmodulin antagonism. I strongly recommend publication in Magnetic Resonance.

The authors may consider the following minor issues:

Apparently, there's only a single set of resonances for the ligand without (in Fig S1) an indication for significant exchange broadening (Fig. S1). Can the authors say something about the timescale of reorientation in the binding pockets ?

As the authors note correctly, the presence of intermolecular NOEs of many ligand resonances to the same protein nuclei, widely distributed across each domain, could be indicative of spin diffusion. The main reason for the choice of 100 ms as mixing time is that it was also used in previous studies. I do no mistrust the interpretation of the observed NOE pattern as a consequence of variable orientations of the ligand in the binding pockets, but additional spectra recorded with shorter mixing time might be of interest (not required though) nevertheless and would settle the point. Also, it is stated that "spin-diffusion effects are not dominant between nuclei within the ligand when it is bound in the complex", but no explanation is given.

In the experimental section, recording of a 3D 13C-half filtered-13C-edited NOESY is described, but the experiment is not mentioned or shown in the Results part. If the experiment was in fact used, why was a 15N filter not applied as well ? There should be some overlap of aromatic ligand signals with amide protons of calmodulin. Does w2-13C-half filtered w1-13C-NOESY (line 126) mean 2D w2-13C-half filtered w1-13C-edited-[1H,1H]-NOESY? The original name of the J-based experiment for methionine e-methyl assignment is Long Range 13C-13C (LRCC) correlation.

Fig. 1: I don't understand the numbering scheme for idoxifene. Where are positions 1-6 ? Why are different numbers used for equivalent positions in the rings ?

The values used for indirect referencing of 13C and 15N (line 109) according to Wishart et al. refer to internal DSS, but TSP is used here instead.

*(2) Authors Response*

We would like to thank the reviewer for taking the time to read and comment on our manuscript, and for their positive reaction to it. Regarding the minor issues raised:

We had included the point - that the number and linewidth of the bound idoxifene resonances in Fig S1 require that the reorientation of CaM-bound idoxifene molecules must be fast relative to the overall dissociation rate - in the Discussion section (on line 415) in the original manuscript. Nonetheless, we have amended the manuscript to better emphasise this point.

We agree that the use of a range of mixing times would allow the extent of any spin-diffusion effects to be quantified thoroughly, but unfortunately we are unable to run any more such experiments on a reasonable timescale, as we no longer have access to idoxifene. The conclusion that spin-diffusion effects are not dominant in the bound ligand comes from the absence of NOEs between protons separated by more than 5 Angstroms in idoxifene. We have amended the manuscript to incorporate this point.

On re-reading the Resonance and NOE Assignment part of the Materials and Methods section we see that it had not been finished properly and hence was quite confusing. We have re-written this section more clearly in the revised manuscript, and harmonized it better with the results section.

The numbering system of idoxifene is derived from previous tamoxifen analogues, and every atom is required to be represented uniquely in the topology file for the structure calculation. Only carbon atoms bonded to H are annotated in Fig. 1. Atoms 1-3 are heteroatoms (I, O, N) and carbons 4, 5, 6, 14, 17, 20, and 23 have no directly bonded H atoms.

We have used the 1H signal of TSP in the referencing process and ignored the up to 0.015 ppm potential discrepancy with the 1H signal of DSS. The chemical shift changes reported will be unaffected, and the referencing is all internally consistent.

*(3) Author Changes*

Line 111: 2.3.1 Resonance and NOE Assignment, replaced text with:

Resonance assignment of free CaM was based on the assignment of *Drosophila* CaM (BMRB entry 547) and it was verified using TOCSY and NOESY-HSQC experiments (Craven et al., 1996). For the idoxifene:CaM complex, the amide $^1$H-$^{15}$N correlations were followed in the titration series of HSQC spectra using 3D TOCSY-HSQC and NOESY-HSQC experiments. The assignments of backbone and sidechain resonances of $^{13}$C,$^{15}$N-labeled CaM in the 2:1 idoxifene:CaM complex were confirmed using previously described protocols (Craven et al., 1996; Osawa et al., 1998), including 3D $^{13}$C-edited NOESY-HSQC and 3D Long Range $^{13}$C-$^{13}$C Correlation experiments for the assignment of the $\varepsilon$ methyl resonances of methionine. The assignment of idoxifene in the absence of protein (Fig. S1) was carried out in D$_2$O at pH 3.0, as the ligand is insoluble at pH 6.0. Idoxifene resonances in the complex were assigned using 2D TOCSY and a 2D $^{13}$C-$^{15}$N-double-half filtered NOESY experiment acquired with a mixing time of 100 ms, in line with previous solution structures of CaM-small ligand complexes (Craven et al., 1996; Osawa et al., 1998). The identity of resonances involved in intermolecular NOEs in the complex was also discerned using the 2D $^{13}$C-$^{15}$N-double-half filtered NOESY experiment and, for protein resonances, confirmed using a 3D $\omega_1$-$^{13}$C-$^{15}$N-half-filtered $^{13}$C-edited NOESY-HSQC spectrum with the same mixing time. For matching to the assignment data, tolerances of 0.03 ppm and 0.3 ppm were used for $^1$H and $^{13}$C frequencies respectively. The chemical shift values of the peak centres were converted to X-PLOR restraints using in-house software. For the idoxifene:CaM complex, 180 ligand-protein NOEs were observed, of which 110 were unambiguously assigned.

Line 248: Corrected: Long Range $^{13}$C-$^{13}$C Correlation

Line 294: 3.3 Structure Determination, now starts:
Intramolecular NOEs within CaM, within idoxifene, and intermolecular NOEs between idoxifene and CaM were quantified using a 2D $^{13}$C-$^{15}$N-double-half-filtered NOESY spectrum of a 2:1 idoxifene:CaM solution (Fig. 5), and the identity of protein resonances was confirmed using a 3D $\omega_1$-$^{13}$C-$^{15}$N-half-filtered $^{13}$C-edited NOESY-HSQC spectrum.

Line 306: inserted ", since NOEs over extended distances are not observed"

Line 316: corrected figure legend

Line 452: inserted "i.e. the exchange rate between sites must be significantly faster than the dissociation rate."

---

## Author Response (AR2)

Comments to the Author:
The authors have addressed the comments by the two referees very well. This is an interesting study reporting an unusual ligand binding mode. It is very well suited for the Rob Kaptein Festschrift. The paper is essentially ready for publication, after resolving a few editorial formatting requests. In addition, please also address the following minor revision.

- The overall finding that the bound ligand populates a conformational ensemble in the hydrophobic pocket of the protein strongly resembles the ligand interaction mode described in Takeuchi et al, Scientific Reports 4, 6922 (2014). Please cite that paper and briefly compare the two cases in the discussion.

We have cited this interesting and relevant paper, and included a sentence relating it to our study in the discussion section. Essentially the study by Takeuchi and co-workers focusses on dynamics within the protein, and ours on dynamics within the ligand.

Non-public comments to the author:
Please make the following formatting changes:

- Figure 2 A, B, C: y-axes need a proper label including units (probably Fluorescence intensity (a.u.) and [CaM]/[idoxifene])

All done

- Figures 3, S2 label both axes, with property and unit.

All done. Note y-axis of Fig 3 and the x-axes of both Figs are unitless.

- Figures 2C, 4, 5, S1, please label axes that shows chemical shifts with labels like delta(1H) [ppm] or omega_X(1H) [ppm], where X is the dimension number.

All done

- Please doublecheck all Figures and Supplementary Figures for proper axes labeling. If self-explanatory, the description of these labels may then be removed from the respective Figure captions to make captions more concise.

We have shortened the caption to Fig 2 accordingly, and haven't spotted any other issues relating to axes labelling.